# Long-Term Baseflow Responses to Projected Climate Change in the Weihe River Basin, Loess Plateau, China

Junlong Zhang [1], Panpan Zhao [2,*], Yongqiang Zhang [3], Lei Cheng [4,5], Jinxi Song [6], Guobin Fu [7], Yetang Wang [1], Qiang Liu [8], Shixuan Lyu [1,9], Shanzhong Qi [1], Chenlu Huang [6], Mingwei Ma [2] and Guotao Zhang [10]

1 College of Geography and Environment, Shandong Normal University, Jinan 250358, China
2 College of Water Resources, North China University of Water Resources and Electric Power, Zhengzhou 450045, China
3 Key Laboratory of Water Cycle and Related Land Surface Processes, Institute of Geographic Sciences and Natural Resources Research, Chinese Academy of Sciences, Beijing 100101, China
4 State Key Laboratory of Water Resources and Hydropower Engineering Science, Wuhan University, Wuhan 430072, China
5 Hubei Provincial Collaborative Innovation Center for Water Resources Security, Wuhan 430072, China
6 Shaanxi Key Laboratory of Earth Surface System and Environmental Carrying Capacity, College of Urban and Environmental Sciences, Northwest University, Xi'an 710127, China
7 CSIRO Land and Water, Wembley, WA 6913, Australia
8 State Key Laboratory of Water Environment Simulation, School of Environment, Beijing Normal University, Beijing 100875, China
9 Department of Civil Engineering, University of Bristol, Bristol BS8 1TR, UK
10 Key Laboratory of Land Surface Pattern and Simulation, Institute of Geographic Sciences and Natural Resources Research, Chinese Academy of Sciences (CAS), Beijing 100101, China
* Correspondence: zhaopanpan@ncwu.edu.cn

**Abstract:** Climate change is a significant force influencing catchment hydrological processes, such as baseflow, i.e., the contribution of delayed pathways to streamflow in drought periods and is associated with catchment drought propagation. The Weihe River Basin is a typical arid and semi-arid catchment on the Loess Plateau in northwest China. Baseflow plays a fundamental role in the provision of water and environmental functions at the catchment scale. However, the baseflow variability in the projected climate change is not well understood. In this study, forcing meteorological data were derived from two climate scenarios (RCP4.5 and RCP8.5) of three representative general circulation models (CSIRO-Mk3-6-0, MIROC5, and FGOALSg2) in CMIP5 and then were used as inputs in the Soil and Water Assessment Tool (SWAT) hydrological model to simulate future streamflow. Finally, a well-revised baseflow separation method was implemented to estimate the baseflow to investigate long-term (historical (1960–2012) and future (2010–2054) periods) baseflow variability patterns. We found (1) that baseflow showed a decreasing trend in some simulations of future climatic conditions but not in all scenarios ($p < 0.05$), (2) that the contribution of baseflow to streamflow (i.e., baseflow index) amounted to approximately 45%, with a slightly increasing trend ($p \leq 0.001$), and (3) an increased frequency of severe hydrological drought events in the future (2041–2053) due to baseflows much lower than current annual averages. This study benefits the scientific management of water resources in regional development and provides references for the semi-arid or water-limited catchments.

**Keywords:** baseflow; Weihe River Basin; Loess Plateau; climate change; General Circulation Models

## 1. Introduction

Distinguishing the contributors of different streamflow components is vital to the effective management of catchment water resources. Baseflow is the contribution of delayed pathways to stream discharge that maintains streamflow during drought periods,

characterized by low precipitation, the dominance of groundwater discharge and/or snow meltwater from upstream regions [1–3]. Baseflow influences the water quality/supply and the health of the catchment ecosystem in regional development [4]. It has a profound influence on the hydrologic cycle in prolonged dry periods [5–7]. It is essential for the provision of water resources and water security that can be influenced by climate conditions [8–10]. Therefore, estimating projected baseflow drought is critical to escalating our understanding of hydrological processes in the changing climate.

Climate variability is the primary factor influencing the terrestrial hydrologic cycle (e.g., baseflow) at regional and global scales [11–14]. For example, baseflow has a close link with the redistribution of precipitation due to infiltration providing a vital contribution to groundwater flow [15], which is characterized by a close interaction between groundwater and surface water. Trancoso, et al. [2] showed that reduced precipitation diminished baseflow and precipitation and positively affected baseflow in eastern Australia. However, the land-surface air temperature has increased over the past three decades and led to an energized/accelerated hydrological cycle by influencing precipitation amounts [16,17] and by capturing longwave radiation [13]. Li, et al. [18] used an analytical approach that integrated water balance and the Budyko hypothesis (evaporative index (ET/P) and aridity index (PET/P) were used to describe the long-term water and energy balance [19,20]) to separate the contributions of climate and anthropogenic effects on streamflow. Li, et al. [21] investigated the response of baseflow to climate variability in a large forested catchment and found that the contribution of climate variability to annual baseflow were greater than the impacts from forest disturbance. Trancoso, et al. [2] predicted a decreasing baseflow trend under certain climate changes (e.g., decreasing precipitation and increasing evapotranspiration related to $CO_2$–vegetation feedbacks). Ficklin, et al. [16] assessed the impacts of climate change on baseflow and stormflow and found that baseflow had consistent trends with stormflow across the northeastern and southwestern United States. Additionally, Singh, et al. [4] quantified the response of baseflow levels to climate variability cycles (e.g., the Pacific Decadal Oscillation) in the Flint River.

Hydrological models are often used to estimate the effects of climatic factors on water yield. Climate projections have predicted that the frequency and intensity of extreme events (e.g., droughts and floods) will increase under future climate conditions [22]. However, the direct consequences of baseflow responses to future climate change are poorly understood. Therefore, assessing baseflow responses under climate change is imperative to facilitate the understanding of groundwater-related hydrological processes and provides scientific guidelines for water adaptation measures [23] in water-limited regions to face future droughts.

Generally, this approach uses alternative emission scenarios to investigate hydrological responses to climate change [24,25]. For instance, Yang, et al. [26] used 16 climate models from CMIP5 (the fifth phase of the Coupled Model Intercomparison Project) to assess the responses of hydrologic drought/aridity to climate change. They demonstrated that climate models did not capture vegetation water use under elevated $CO_2$ conditions. Semi-distributed rainfall-runoff models based on SWAT (Soil & Water Assessment Tool, https://swat.tamu.edu/, accessed on 9 July 2016) have been widely used to evaluate streamflow variations in complex catchments [27]. Zhang, et al. [28] compared the performances of two distributed hydrological models (e.g., SWAT and the Distributed Hydrology Soil Vegetation Model) in separating the impacts of climate change and LUCC (land-use cover change) on catchment hydrology. Lauffenburger, et al. [29] evaluated the effects of agricultural irrigation and future climate change on groundwater recharge in the northern High Plains aquifer, USA, and found a significant bidirectional shift, leading to a reduction in future groundwater recharge. While those efforts improved our understanding of climate-variability effects on hydrological processes, baseflow responses to future climate change are poorly understood for semi-arid catchments in loess deposition regions, in which baseflow provides a significant water source for ecological restoration and environmental protection.

The Weihe River Basin (WRB) is a representative catchment on the Loess Plateau. It is one of the most important water sources for the environment and regional society of northwest China. In this study, to attenuate the uncertainties of baseflow estimation (e.g., signal and magnitude [30]), historical daily streamflow data and future streamflow data projected by two climate scenarios from three presentative GCMs were used to assess temporal variations in baseflow and the dynamics of baseflow characteristics under future climate changes in the WRB. The specific objectives were (1) projecting baseflow under two scenarios (RCP4.5 representing a lower emissions scenario, and RCP8.5 representing a higher emission scenario) from three GCMs (CSIRO-Mk3-6-0, MIROC5, and FGOALSg2); (2) assessing baseflow responses under future climate conditions; and (3) highlighting the role of baseflow in drought events at the catchment scale. Thus, this study provides drought assessment for water-resource managers to face the future changing climate.

## 2. Study Area and Data Sources

### 2.1. Study Area Description

The Weihe River has a total length of 818 km and is located in the northern Qinling Mountains. It is the largest tributary of the Yellow River. The WRB covers three terrain sections, i.e., the Loess Plateau, the Guanzhong Plain, and the Qinling Mountains, and spans $6.72 \times 10^4$ km$^2$ from north to south (Figure 1). The Weihe River has its source at Niaoshu Mountain (3485 m) in the Gansu Province, flows from west to east, and joins the main channel of the Yellow River in Tongguan County. The longitudinal inclination of the river is about 1.7‰ [31], and the lowest and highest elevations are 325 and 3485 m [32], respectively.

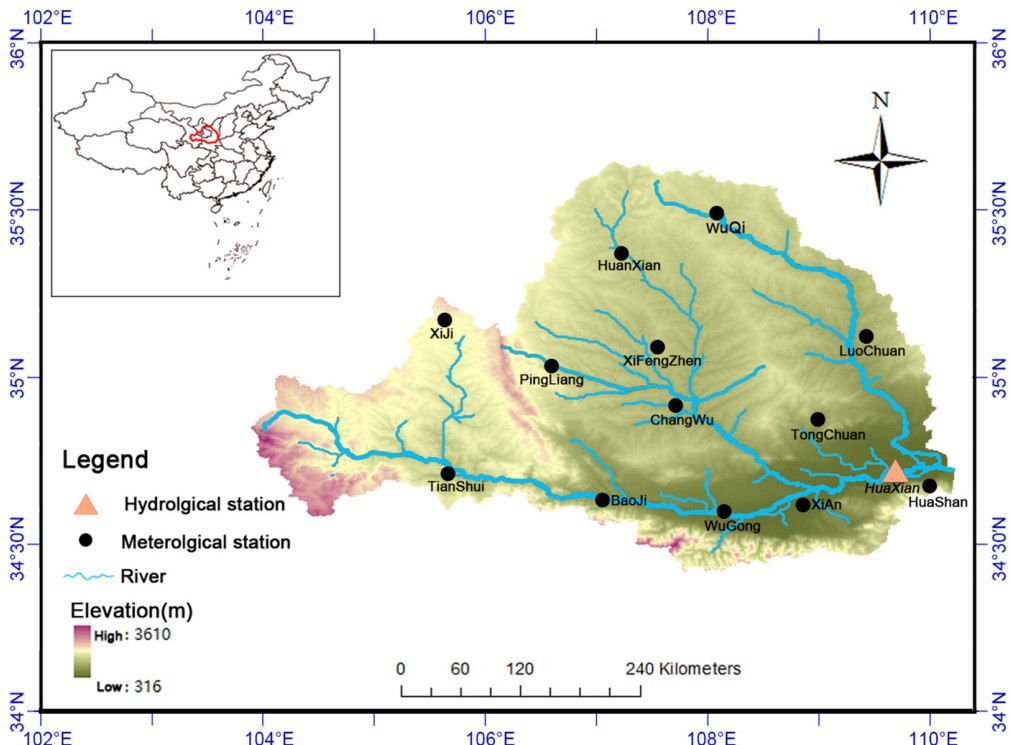

**Figure 1.** Study area, hydrological and meteorological stations in this study.

The climate of this basin is characterized by the continental monsoon with cold, dry, and rainless winters; hot and rainy summers [33]; average annual temperature changes between 7.8 and 13.5 °C; and annual precipitation between 558 and 750 mm [34]. The seasonal distribution of precipitation is uneven, and high precipitation and flow mainly occur in flood periods (June to September). Both precipitation and runoff have substantial

inter-annual and intra-annual variabilities. The mean annual potential evaporation is approximately 800 mm in the south to 1200 mm in the north [35].

The WRB has extensive loess deposits across the mid and northern catchment. Its soil has a relatively high infiltration potential, and its southern part is primarily covered by forested land in the Qinling Mountains. The predominant land use is agricultural (i.e., wheat and cotton production [36]) in the center of the basin, where cultivated soils have been subjected to long-term agricultural development. Cultivated land covers more than 50% of the basin, followed by woodland and grassland [33]. The basin is highly productive and supplies water and food for the region. However, streamflow and groundwater have decreased rapidly with historical increases in population, agricultural production, industries, and related developmental activities [35]. Land-use changes, particularly due to the large ecological plan (e.g., the Grain for Green Program [37,38]) launched in the 1990s, have significant impacts on the catchment's hydrology [1,39].

### 2.2. Data Sources

Daily precipitation data, covering 1960 to 2012, from the 13 standard meteorological stations (Table 1) in the WRB were obtained from the China Meteorological Administration (http://cdc.cma.gov.cn, accessed on 22 August 2015). These meteorological stations are maintained according to the standard methods of the National Meteorological Administration of China. For the same period, daily streamflow data from the Huaxian gauge (Figure 1) was obtained from the Hydrological Yearbooks of China (http://loess.geodata.cn, accessed on 10 May 2016). All meteorological and hydrological data used in this study have been submitted to quality control by government agencies before release.

**Table 1.** Meteorological stations used in this study.

| Station ID | Station | Latitude | Longitude | Elevation (m) |
|---|---|---|---|---|
| 53738 | WuQi | 36.95 | 108.17 | 1331.4 |
| 53821 | HuanXian | 36.58 | 107.3 | 1255.6 |
| 53903 | XiJi | 35.97 | 105.78 | 1916.5 |
| 53915 | PingLiang | 35.55 | 106.57 | 1346.6 |
| 53923 | XiFengZhen | 35.73 | 107.63 | 1421 |
| 53929 | ChuangWu | 35.2 | 107.8 | 1206.5 |
| 53942 | LuoChuan | 35.82 | 109.5 | 1159.8 |
| 53947 | TongChuan | 35.08 | 109.07 | 978.9 |
| 57006 | TianShui | 34.58 | 105.75 | 1141.7 |
| 57016 | BaoJi | 34.35 | 107.13 | 612.4 |
| 57034 | WuGong | 34.25 | 108.22 | 447.8 |
| 57036 | XiAn | 34.3 | 108.93 | 397.5 |
| 57046 | HuaShan | 34.48 | 110.08 | 2064.9 |

## 3. Methods

### 3.1. Baseflow Separation Algorithm

To improve the accuracy of baseflow estimates in this study, revised and validated baseflow separation was implemented [40]. Baseflow has a lag time concerning the last precipitation event [41]. Generally, the baseflow recession is linked with the surface and sub-surface flow characteristics and follows an exponential decay curve [42]:

$$Q_b = Q_0 \alpha^t \tag{1}$$

where $Q_b$ is the baseflow at time $t$, and $\alpha$ is the recession constant determined by recession analysis. The baseflow can be calculated using the baseflow separation method.

Baseflow separation is a fundamental issue that has been comprehensively documented [8,43,44]. Several algorithms have been proposed to separate baseflow from total observed streamflow [45–47] and can be classified as trace-based, water balance, and graph approaches according to general applications. Digital filters are the most widely used tools for small-data input and is reducible (e.g., only daily streamflow records and more objective) [40]. The Lyne–Hollick method was used here, expressed as [48]:

$$Q_{q\ (i)}\ =\ \alpha Q_{q\ (i\ -\ 1)}\ +\ \frac{1\ +\ \alpha}{2}(Q_i\ -\ Q_{i\ -\ 1}) \qquad (2)$$

where $Q$ is total streamflow (m$^3$/d), $Q_q$ is quick flow (mm/d), $i$ is the time step (day), and $\alpha$ is the filter parameter (recession constant, in 1/day). Baseflow ($Q_b$, m$^3$/d) can subsequently be calculated as $Q_i$ minus $Q_q$. The baseflow index (BFI, calculated as *total $Q_b$/total Q*), is a standard indicator of the baseflow contribution to total streamflow. Herein, the calibrated Lyne–Hollick method was employed to separate the long-term baseflow. This approach has been validated by Zhang, et al. [40].

The recession constant can be obtained using the recession analysis developed by Brutsaert, et al. [43]. This recession approach efficiently reduces uncertainties when estimating the initial points in the recession limb. Details of recession analysis are given in Cheng, et al. [44].

### 3.2. Selection of General Circulation Models

The general circulation model (GCM) is widely used to estimate the impacts of future climate conditions on hydrological cycles [26,49–52]. The GCMs used in this study (Table 2) were available in the Intergovernmental Panel on Climate Change (IPCC) data archive (https://pcmdi.llnl.gov/mips/cmip5/, accessed on 16 October 2016). Based on monthly precipitation data from 40 GCMs for two representative concentration scenarios (RCP4.5 and RCP8.5) and the future climate scenario period based on CMIP5, we divided GCM data into two sections. The 45 years from 1960–2004 (historical climate period, HCP) were considered the baseline period, and the 45 years from 2010–2054 were the future climate period (FCP).

**Table 2.** Summary of 40 general circulation models (GCM) selected in this study.

| ID | GCM | Originating Group (s) | Country | Resolution (°) |
|----|-----|-----------------------|---------|----------------|
| 1 | ACCESS1.0 | CSIRO-BOM | Australia | 1.88 × 1.25 |
| 2 | ACCESS1.3 | CSIRO-BOM | Australia | 1.88 × 1.25 |
| 3 | BCC-CSM1.1 | BCC | China | 2.81 × 2.81 |
| 4 | BCC-CSM1.1.M | BCC | China | 1.13 × 1.12 |
| 5 | BNU-ESM | BNU-ESM | China | 2.81 × 2.81 |
| 6 | CanESM2 | CCCMA | Canada | 2.81 × 2.79 |
| 7 | CCSM4 | NCAR | USA | 1.25 × 0.94 |
| 8 | CESM1(BGC) | NCAR | USA | 1.25 × 0.94 |
| 9 | CESM1(CAM5) | NCAR | USA | 1.25 × 0.94 |
| 10 | CESM1(WACCM) | NCAR | USA | 2.5 × 1.89 |
| 11 | CMCC-CM | CMCC | Italy | 0.75 × 0.75 |
| 12 | CMCC-CMS | CMCC | Italy | 1.88 × 1.88 |
| 13 | CNRM-CM5 | CNRM-CERFACS | France | 1.41 × 1.40 |
| 14 | CSIRO-Mk3.6.0 | CSIRO-QCCCE | Australia | 1.88 × 1.88 |
| 15 | EC-EARTH | MOHC | UK | 1.13 × 1.13 |
| 16 | FGOALS-g2 | LASG-GESS | China | 2.81 × 3.05 |
| 17 | FGOALS-s2 | LASG-IAP | China | 2.81 × 1.41 |
| 18 | FIO-ESM | FIO | China | 2.81 × 2.81 |
| 19 | GFDL-CM3 | NOAA GFDL | USA | 2.50 × 2.00 |
| 20 | GFDL-ESM2G | NOAA GFDL | USA | 2.50 × 2.00 |

**Table 2.** *Cont.*

| ID | GCM | Originating Group (s) | Country | Resolution (°) |
|----|-----|------------------------|---------|----------------|
| 21 | GFDL-ESM2M | NOAA GFDL | USA | 2.50 × 2.00 |
| 22 | GISS-E2-H | NASA GISS | USA | 2.50 × 2.00 |
| 23 | GISS-E2-H-CC | NASA GISS | USA | 2.50 × 2.00 |
| 24 | GISS-E2-R | NASA GISS | USA | 2.50 × 2.00 |
| 25 | GISS-E2-R-CC | NASA GISS | USA | 2.50 × 2.00 |
| 26 | HadGEM2-AO | KMA/NIMR | UK/Korea | 1.88 × 1.25 |
| 27 | HadGEM2-CC | KMA/NIMR | UK/Korea | 1.88 × 1.25 |
| 28 | HadGEM2-ES | KMA/NIMR | UK/Korea | 1.88 × 1.25 |
| 29 | INMCM4 | INM | Russia | 2.00 × 1.50 |
| 30 | IPSL-CM5A-LR | IPSL | France | 3.75 × 1.89 |
| 31 | IPSL-CM5A-MR | IPSL | France | 2.50 × 1.27 |
| 32 | IPSL-CM5B-LR | IPSL | France | 3.75 × 1.89 |
| 33 | MIROC5 | MIROC | Japan | 1.41 × 1.40 |
| 34 | MIROC-ESM | MIROC | Japan | 2.81 × 2.79 |
| 35 | MIROC-ESM-CHEM | MIROC | Japan | 2.81 × 2.79 |
| 36 | MPI-ESM-LR | MPI-M | Germany | 1.88 × 1.87 |
| 37 | MPI-ESM-MR | MPI-M | Germany | 1.88 × 1.87 |
| 38 | MRI-CGCM3 | MRI | Japan | 1.13 × 1.12 |
| 39 | NorESM1-M | NCC | Norway | 2.50 × 1.89 |
| 40 | NorESM1-ME | NCC | Norway | 2.50 × 1.89 |

In the context of climate change, three GCMs (i.e., the dry, moderate, and wet effects) were chosen to represent the future climate conditions. Then, daily precipitation and temperature derived from GCMs were used as forcing data to project streamflow in the FCP. The representativeness of the ensemble GCMs is considerably improved in the projection of climate variables [53]. Among the 40 GCMs under the two scenarios in CMIP5, the numbers of GCMs predicting increasing and decreasing future precipitation were 36 and 4, respectively. To choose representative models and reduce uncertainties, three models were selected to simulate future climate conditions, i.e., CSIRO-Mk3-6-0 (predicting dry conditions with the largest precipitation declines), MIROC5 (wet conditions with the largest precipitation increases), and FGOALSg2 (median conditions with a median change in precipitation) (Figure 2).

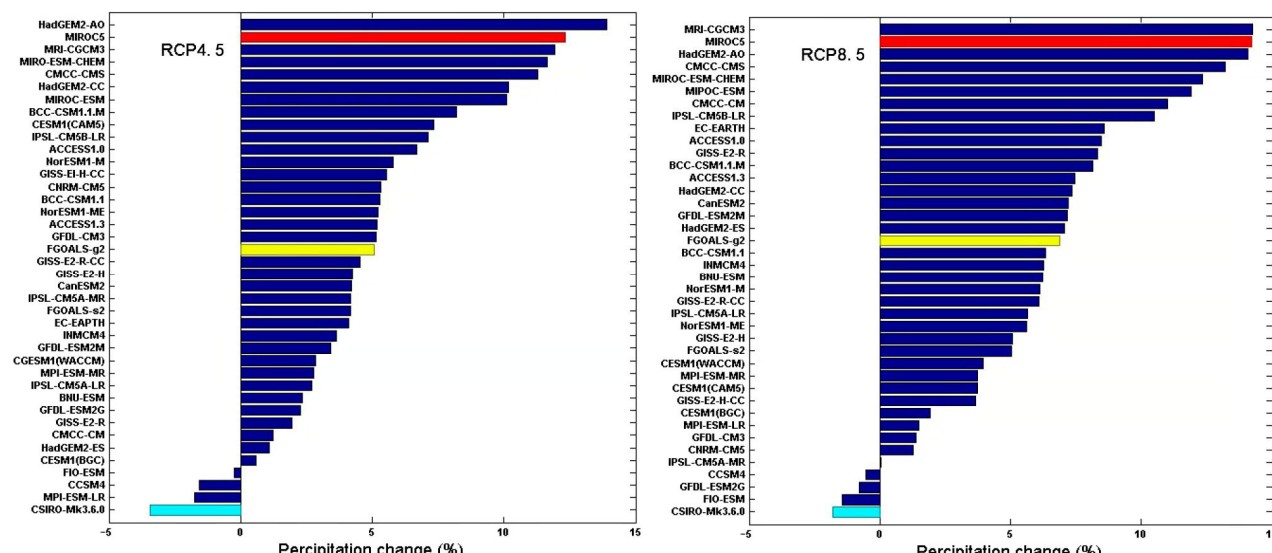

**Figure 2.** General circulation model selection of the dry, moderate, and wet simulated effects for two scenarios (RCP4.5 and RCP8.5).

To generate the mean climate conditions, the GCMs' climate projections were bias-corrected with the delta-change method (for details, see Navarro-Racines, et al. [54]), which simply superimpose the mean monthly anomalies between the GCMs-simulated baseline and the future period on the observed historical precipitation and temperature to represent future climate. Specifically, first, we calculated the ratio between the observed and simulation precipitation data of the three selected GCMs in the historical period (1960–2004). Second, we multiplied or added the precipitation and temperature data of the three GCMs in the future period (2010–2054) with this ratio to obtain simulation data for the FCP. Finally, we used the simulation data as forcing input data for running SWAT (Soil and Water Assessment Tool) to estimate daily streamflow.

### 3.3. SWAT Model

The SWAT hydrological model is a continuous-time, computationally efficient, and semi-distributed catchment-scale hydrologic model [55]. The catchment was divided into hydrological response units (HRUs), and surface runoff volumes were simulated for each HRU. SWAT has been widely used in different catchments worldwide and proved to be an effective tool to examine hydrological responses to land use and climate changes [56]. More details on SWAT are given in Easton, et al. [57], Guo, et al. [58].

This study used daily meteorological data (precipitation, maximum and minimum temperature, mean wind speed, radiation, mean relative humidity) from 1960–2012 as forcing data to simulate daily runoff in the WRB. The performance of predicted runoff was assessed against observed daily streamflow data in the same period. In the SWAT simulation, 1983–2012 was the calibration period (warm-up period: 1983–1993), and 1960–1982 was the validation period (warm-up period: 1960–1969). Comparing the simulated runoff between the calibration and validation period, the simulation of monthly runoff using the SWAT model had a good performance in WRB (Figure 3).

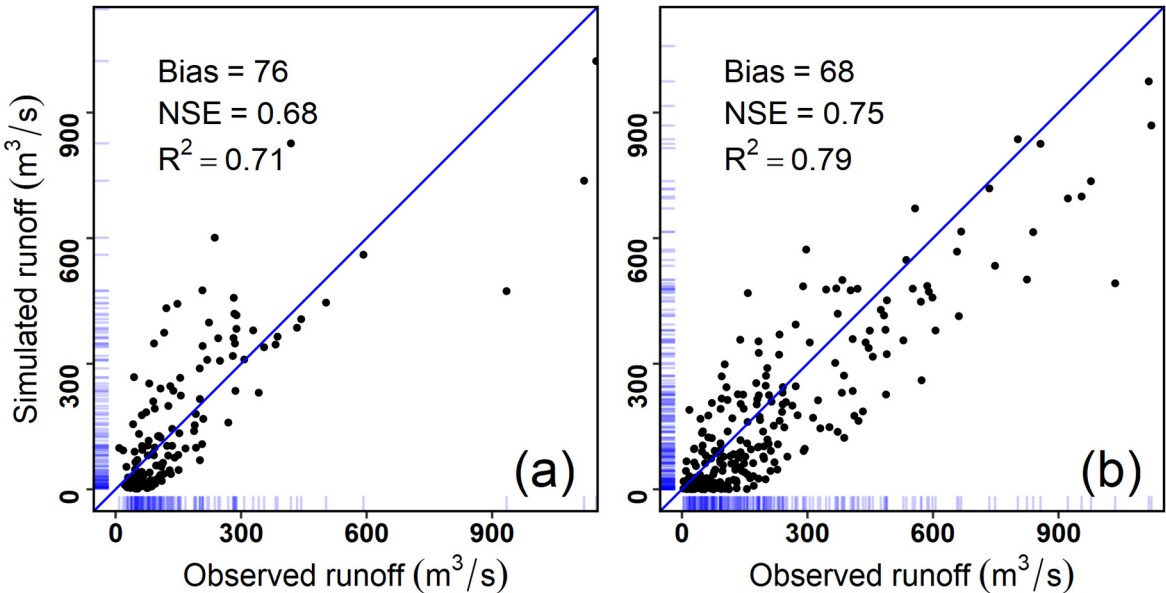

**Figure 3.** The streamflow simulation using SWAT model in calibration (**a**) and validation (**b**). The blue line is the 1:1 line. The rug represents the data distribution density.

### 3.4. Trend Analysis

Trend analysis can provide effective and useful information on possible tendencies in the future [59]. The nonparametric Mann Kendall test was used to identify trends and trend significance in baseflow in this study. This test provides two parameters, i.e., the significance level and slope magnitude [60]. $p$ values $\leq 0.05$ were considered significant.

The Z (derived from a certain climate element sequence) and S are the trend and order column and are used to detect the significance test. This test method has been widely employed to detect significant monotonic increasing or decreasing trends in long-term time-series data [8]. Method details can be found in previous studies [61,62].

*3.5. Baseflow Drought Determination*

Due to the hydrological drought with a higher accumulation period [63] and to provide insights for water planning and drought alerts for other basins facing water shortage events, the annual baseflow anomalies were implemented to determine hydrological droughts in historical and future climate conditions.

## 4. Results

*4.1. Baseflow Estimation*

The meteorological outputs in the GCMs were extracted as the inputs to the SWAT model to predict the streamflow in the future climate change, and then, a well-revised Lyne–Hollick method was used to implement the baseflow separation. Overall, the SWAT model had a good performance in both the calibration (Figure 3a) and validation (Figure 3b) stages to simulate the streamflow on a long-term scale (e.g., with $R^2 > 0.7$). In addition, the annual mean baseflow in the calibration and validation of the SWAT model also is shown in Figure 4.

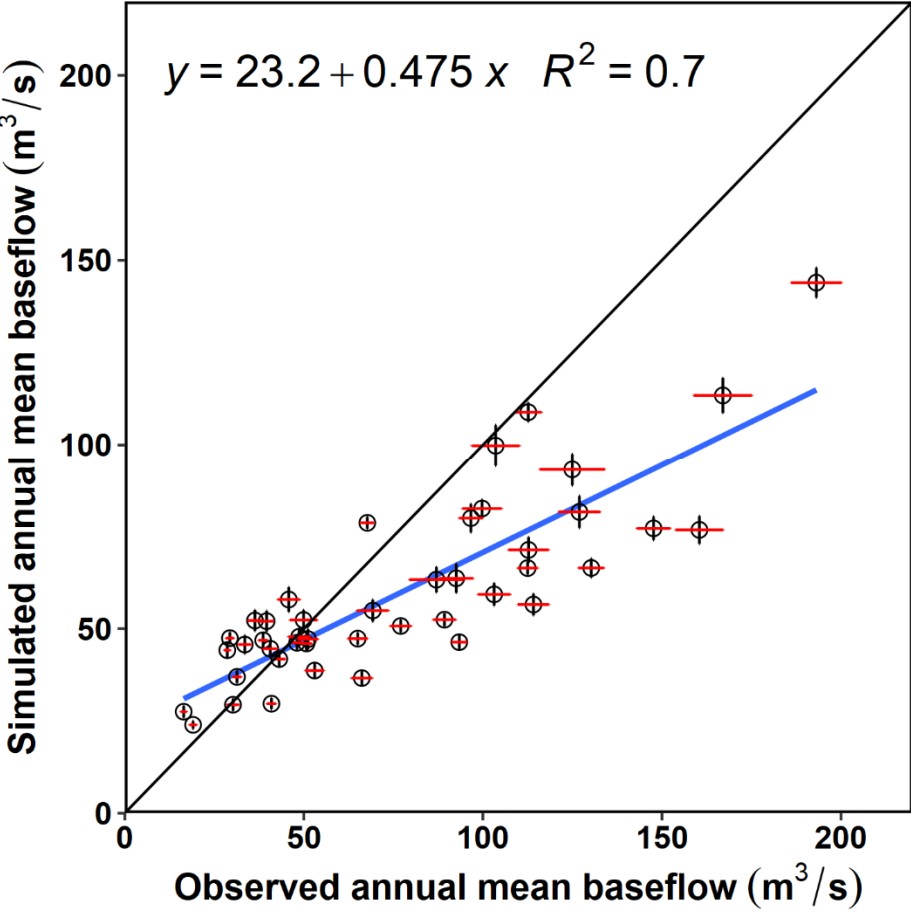

**Figure 4.** Relationship between the simulated and observed annual mean baseflow in the historical period. The black and red error bars represent standard errors in simulated and observed annual total baseflow. The black line is the 1:1 line.

The baseflow time scale of predicted streamflow and recession constants are shown in Table 3. The K and $\alpha$ are ranged from $62 \pm 6$ days (SD) and $0.98 \pm 0.002$ (SD) 1/day

in the two future climate scenarios, respectively. For the CSIRO and FGOALSg2 models, the prediction of streamflow under the two scenarios was very close. For the highest streamflow condition, the MIROCS baseflow results were much higher than those of the other two models.

**Table 3.** Recession analysis derived from three general circulation models and two scenarios for future climate conditions (2010–2054).

| Scenario | GCM | K (days) | $\alpha$ (1/day) |
|---|---|---|---|
| RCP4.5 | CSIRO-Mk3-6-0 | 53.2 | 0.981 |
| | FGOALSg2 | 64.5 | 0.985 |
| | MIROC5 | 69 | 0.986 |
| RCP8.5 | CSIRO-Mk3-6-0 | 54.3 | 0.982 |
| | FGOALSg2 | 67.1 | 0.985 |
| | MIROC5 | 63.7 | 0.984 |

### 4.2. Detection of Baseflow Changes

All three models showed an insignificant increasing trend in both scenarios before 2020 in the future period (Figure 5). From the perspective of changing points, there was a similar and/or general pattern over a long future period; nevertheless, the numbers of changing points were different. In 2020, 2026 and 2034, changing points occurred for CSIRO and FGOALSg2 in both scenarios and for MIROC5 in the RCP8.5 scenario. After 2020, all three models showed an insignificant decreasing trend in both two scenarios. Specifically, in the FGOALSg2 model, there was a significant decreasing trend, and in the MIROC5 model, this trend occurred after 2049.

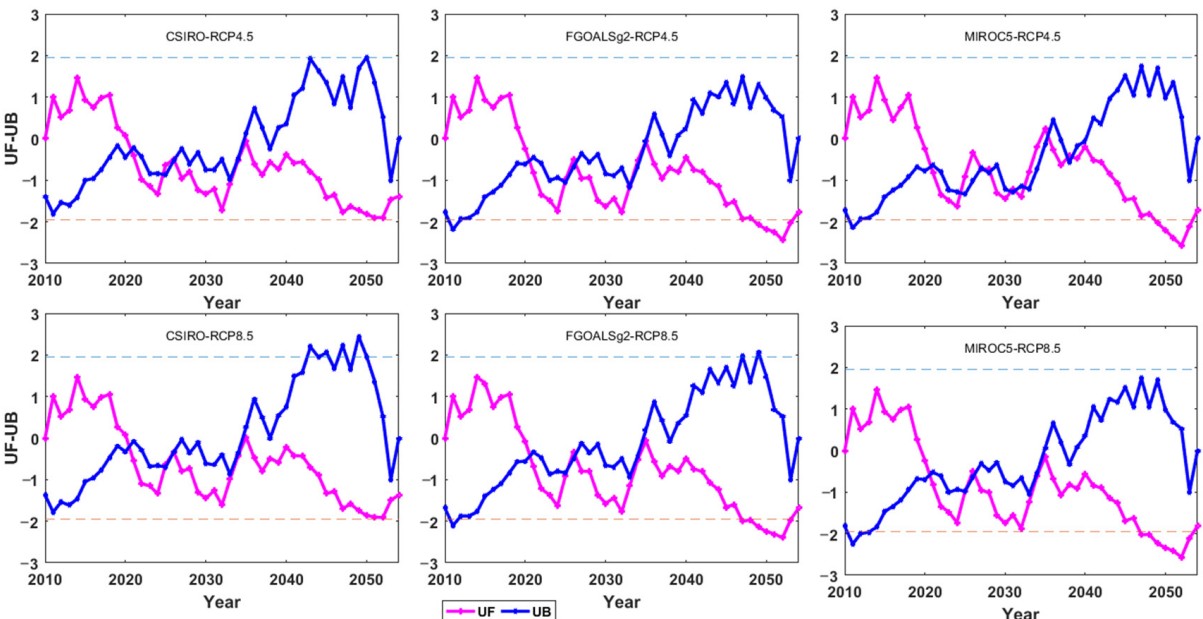

**Figure 5.** Mann Kendall test statistics for three GCMs in two scenarios (RCP4.5 and RCP8.5). UF is the sequential values of a statistic under the random hypothesis; UB is the reversed UF data statistic series. The positive and negative values indicate the increasing and decreasing trend. The intersections of UF and UB present the changing point.

The baseflow derived from the observed daily streamflow (Figure 6) showed a changing point in 1970. Before this year, baseflow showed an insignificant trend. However, after this year, there was a decreasing trend, both in 1977–1983 and after 1995.

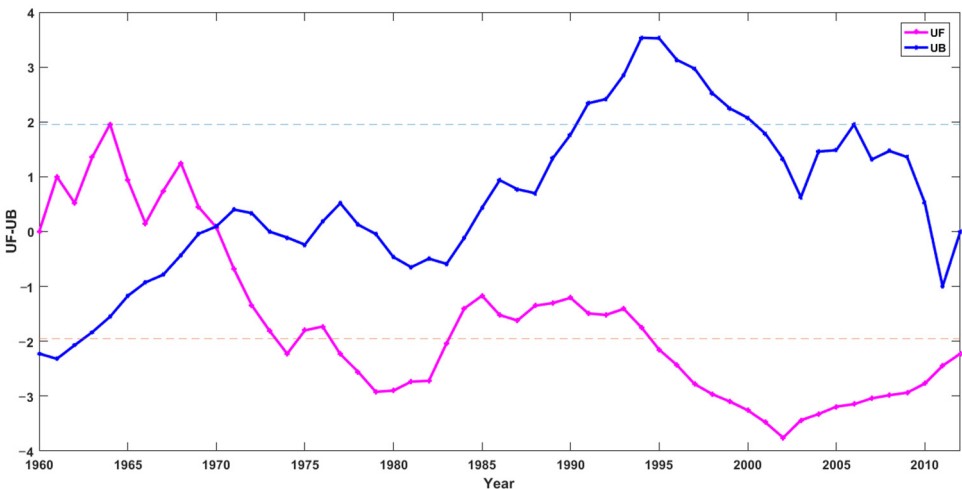

**Figure 6.** Mann　Kendall test statistics for the baseflow separated from historical observed daily streamflow data. The abbreviations are the same as Figure 5.

### 4.3. Quantitative Baseflow Analysis Combining Historical and Future Climatic Conditions

The baseflow exhibited a decreasing trend in the long-term periods (all $p \le 0.005$, see Figure 7). Herein, we first calculated the baseflow anomaly for the entire time series and then added the regression line for each GCM using local polynomial fitting. Specifically, CSIRO had a relatively more variation compared to the other two GCMs in both climate scenarios. Despite the trend with fluctuations, the three GCMs showed a similar performance in the two climate scenarios.

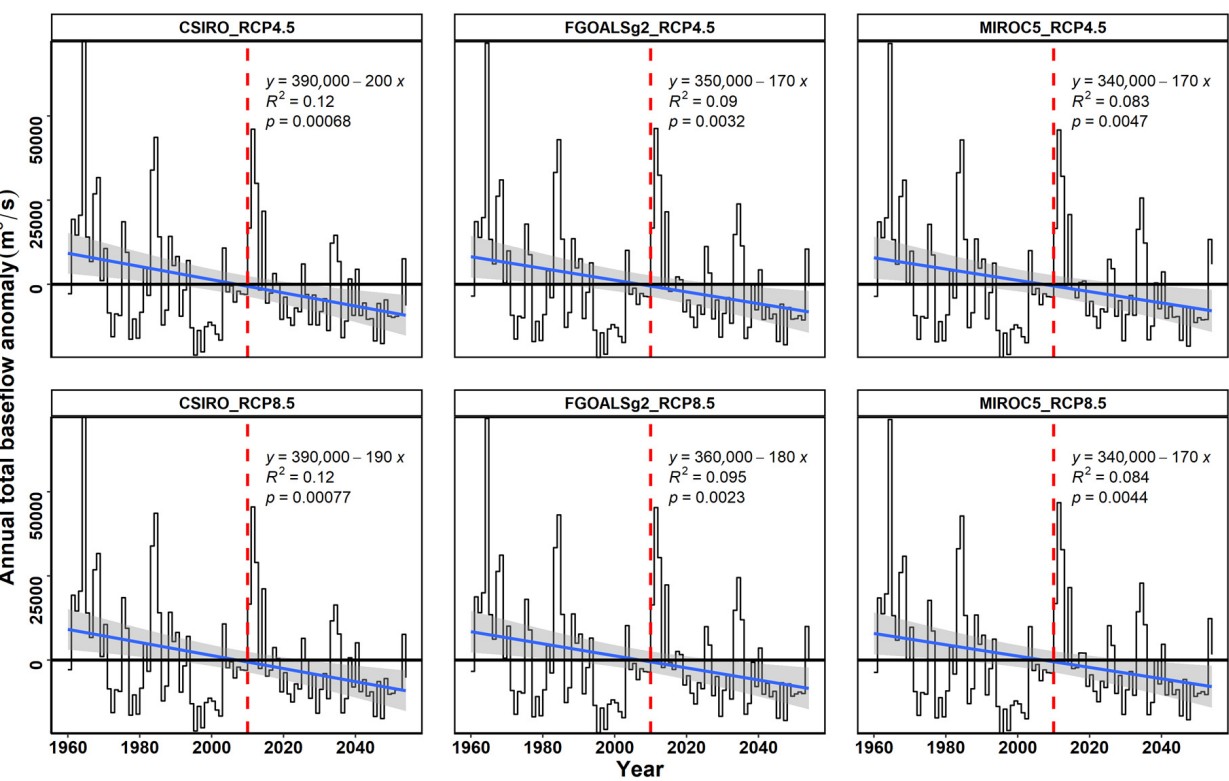

**Figure 7.** Baseflow anomaly plot from the entire streamflow time series in three models and two scenarios. The blue line is the linear regression line. The red vertical dashed line divides the time series into historical and future periods.

## 5. Discussion

### 5.1. Baseflow Trends in Historical and Future Climate Periods

The baseflow separation algorithm used in this study was derived from the revised version of the Lyne–Hollick algorithm. The outcomes of this method were more reproducible than the traditional methods (e.g., graphical approaches and empirical function [40]), thus this approach would greatly reduce the uncertainties of baseflow estimation. As baseflow was not directly measured under experimental conditions and was often estimated from the original total streamflow [40], in addition, the digital filter combined the recession analysis with more physical meanings containing more catchment-specific groundwater drainage characteristics [44] and provides a robust tool to decrease uncertainties in baseflow estimation [64].

The climate scenarios provided a robust tool to project the water balance of the catchment [65], and detecting trend characteristics was beneficial to understanding the hydrological variability at a long-term scale. In this study, the MK test was adopted to detect baseflow changes under future climate conditions (Figures 4 and 5). A declining baseflow trend was predicted for future climate scenarios. The baseflow change point years were 1970 and 1990. These years are not consistent with the streamflow change points reported by Zhan, et al. [33]. It was demonstrated that runoff had a decreasing trend in this basin after 1990 due to human activities, and the changing points of streamflow lagged the baseflow changes by about 20 years. However, the baseflow change points were in the range of the streamflow change points in another catchment on the Loess Plateau. Herein, the streamflow change points for different sub-catchments ranged from 1970 to 1990 [66]. As a delayed water resource, baseflow provides water to the land surface and sustains ecological health under dry spells.

Projections of baseflow and trend analysis are important to prevent and palliate drought losses on the catchment and regional scale [67]. Analyses of climate variability and baseflow improve our understanding of the effects of drought on environmental protection [4]. A drier trend has been reported for most areas of China based on PDIS (the Palmer Drought Severity Index) [67]. The degree of drought is characterized by a high frequency and has a long-term effect on hydrological connections in the WRB. Baseflow characteristics were used to evaluate hydrological droughts because baseflow is relatively steady and can represent catchment water storage [68]. Quantifying the impacts of climate variability on baseflow can provide insights for future water-resources plans [4]. Yang, et al. [41] showed that baseflow recovery had a longer lag than streamflow recovery across 130 unimpaired catchments in eastern Australia. Further, it has been reported that the hydrological cycle is intensified with changes in global mean precipitation in GCM projections [69]. This means that dry areas with limited water may become much drier.

### 5.2. Variability of the Baseflow Index

The BFI is an important hydrological indicator representing the water flow from groundwater/delayed resources to streamflow. It contains a lot of information on catchment characteristics [70,71], which reflects the holistic attribute of baseflow and terrestrial water balance [72]. The relationship between total baseflow, streamflow, and the baseflow index was demonstrated in Figure 8. To address the total baseflow contribution to streamflow, we also assessed the BFI for the historical observed and the simulated results for the three GCMs (Figure 9). There was an increasing trend in the BFI in the long-term climatic period. This means that the role of baseflow was remarkably strengthened in the sustenance of local water in this catchment. Compared to baseflow yield, the BFI is a relative ratio that varied from 0.42 to 0.49 and averaged 0.45 in our study. This means that the contribution of baseflow from groundwater storage or delayed sources accounted for 45% in the WRB from the perspective of future climate conditions in GCM projections. The magnitude of baseflow was very similar in the three models. Nevertheless, streamflow showed relatively greater variations. This confirmed that the baseflow is a relatively stable flow that sustains the terrestrial hydrological ecosystem [73].

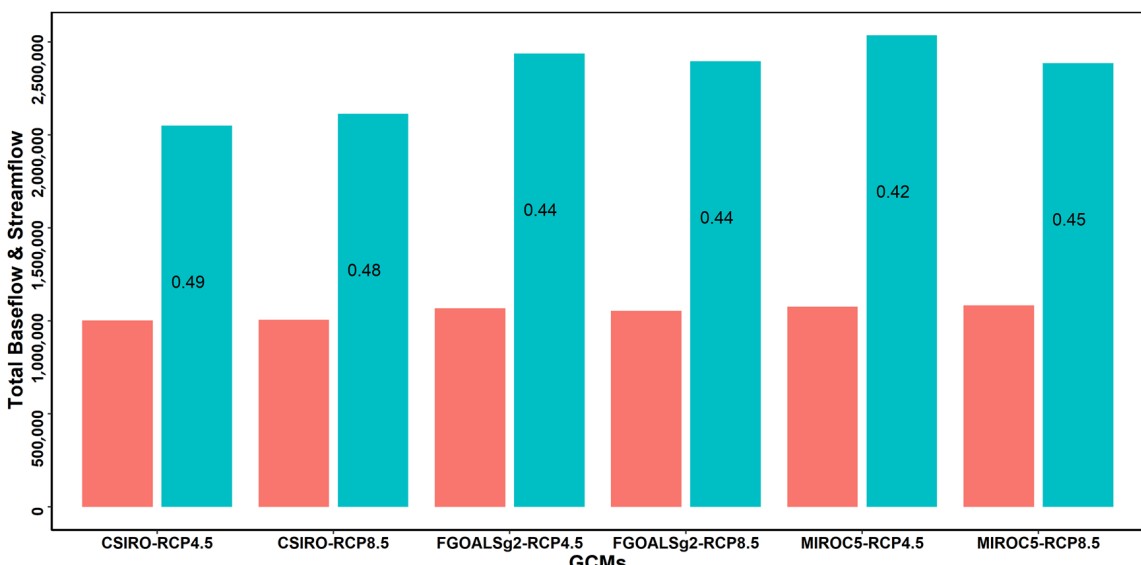

**Figure 8.** Total baseflow and streamflow for each GCM. Numbers in bars are the baseflow index. Red bars are the total baseflow, and blue bars are the total streamflow.

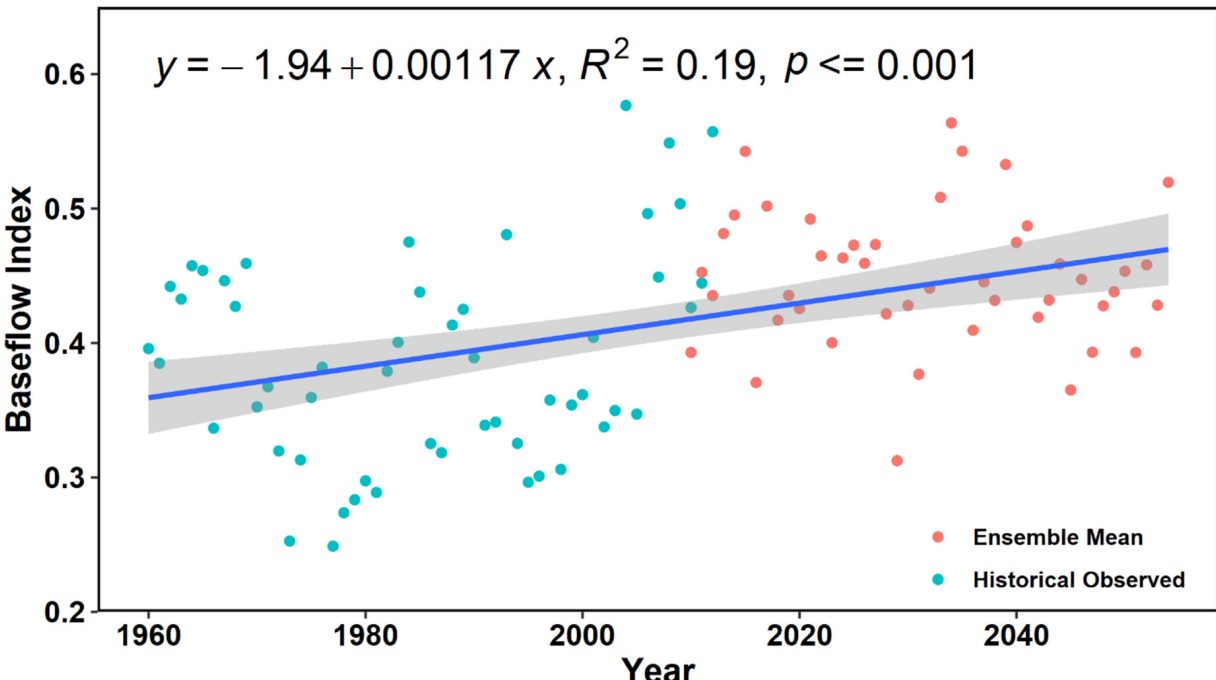

**Figure 9.** Variation in the baseflow index derived from historical observed streamflow data (1960–2010) and the ensemble means of three general circulation models for two scenarios (simulated from six models) for the future climate changes (2010–2054). The line and equation represent a linear regression.

Additionally, to clarify baseflow and streamflow trends, the relationships of historical data and/or projected streamflow and baseflow from the three models under two scenarios were also assessed (Figures 4 and 5). The response of baseflow to streamflow had a relative laggy time interval. This may be related to the increasing degree of anthropogenic activities in this basin due to the heavy exploitation intensity of groundwater resources. Singh, et al. [72] reported that groundwater abstraction significantly influenced flow regimes, with higher baseflow under constrained pumping conditions. Further, the effects of baseflow

increase vary among river reaches, and baseflow and stormflow increases have relatively greater impacts on downstream areas by increasing flow volume [16]. Estimating other anthropogenic pumping effects is a meaningful way to assess the baseflow response to local hydrological variations.

*5.3. Factors Influencing Baseflow Variations*

It has been reported that climate change and anthropogenic activities are drivers of groundwater storage [74,75]. The baseflow yield is associated with the interactions between climate variability and vegetation changes [66,76] and would be influenced by a variety of catchment physical factors [72]. It is characterized by seasonal precipitation variations, i.e., from June to September, which creates the summer-dominated baseflow feature in this basin [1]. Furthermore, land-use change has affected 50% of the area on the Loess Platea [66]. This directly influences streamflow and leads to changes in baseflow. The basin covers three geological classes. Land use was predominantly agricultural in the long term. However, due to the widely distributed loess-deposition areas is in this region, extensive agricultural development causes heavy soil erosion and water-conservation issues [77]. To sustain the water quality and supply of the WRB, the government has taken measures to prevent ecosystem recession (e.g., soil-conservation measures [78]). This should lead to delayed surface runoff and increase the baseflow in small catchments [60]. However, in dry seasons on a long-term scale, the baseflow should be reduced by the loss of groundwater through more plant evapotranspiration. This is associated with vegetation-type changes from grass/bare land to the forested area [79]. Additionally, this complex effect is also influenced by other potential conditions such as topography. For example, Li, et al. [80] showed that the topography plays a paramount role in low flows (flow magnitudes $\leq Q_{75\%}$) in snow-dominated catchments.

The effects of anthropogenic activities associated with agricultural production also strongly control the water cycle in catchments [35]. It has been shown that the plantation intensity on agricultural land reduces downstream water availability [76]. Irrigation is an important factor influencing groundwater processes [29], leading to variations in baseflow [8]. The WRB is the main agricultural region, with large irrigation areas responsible for the food production for the regional population. To maintain living standards and sustain ecological health, the water demands have been increased for decades, and groundwater pumping supports much of the municipal water demand. Additionally, from the perspective of water depletion, agricultural development, and ecological recovery projects were all needing a large amount of water, including surface water and groundwater, it would create a baseflow shortage event for the Loess Plateau. For example, large-scale afforestation may exacerbate baseflow conditions as evapotranspiration increases through the amplification of leaf area and rooting depth [2,81]) for the catchment with constant precipitation input. This impact on baseflow variations would be amplified by climatic variations in this basin.

*5.4. Implications of Baseflow Droughts*

To provide insights for water planning and drought alerts for other basins facing water-shortage events, the annual baseflow anomalies were implemented to determine the hydrological droughts in historical and future climate conditions (Figure 7). It is noted that the baseflow has an apparent decreasing trend overall. Specifically, there was a relatively richer baseflow in approximately 2035. However, there was a lack of baseflow in 2041–2050, leading to a prolonged impact on the hydrological cycle (e.g., baseflow hydrological droughts) in the long term (~10 years).

It has been reported that, when disentangling climatic effects (e.g., precipitation) on hydrology, the uncertainties were much larger in the high-emission scenario RCP8.5 than the relatively low-emission scenario RCP4.5 [52]. In the baseflow estimation in the FCP, there was no remarkable difference between higher- and lower-emission scenarios, and the annual baseflow anomalies were very similar (Figure 7). The uncertainties of this study were likely associated with coarse temporal or spatial resolution and systematic

errors derived from GCMs [67]. Besides, the baseflow relies upon runoff estimates in the model, confined by temperature and precipitation [82]. The variations of the climate phenomenon of wet-getting-wetter and dry-getting-drier [83] also influence baseflow changes in the catchment.

## 6. Conclusions

Climate selection is an important factor influencing hydrological processes. In this study, a physical-based baseflow separation filter was used to separate baseflow from total streamflow to assess baseflow responses to climate (e.g., varying temperature and rainfall). Three representative general climate models with two climate scenarios were used to predict baseflow and analyze trends and driving forces in the Weihe River Basin. Our analyses proved that the GCMs could capture the streamflow variations under future climate conditions and could be used to investigate baseflow characteristics at the basin scale. Our findings showed that the selection of climate had an approximate impact on the baseflow projection. The baseflow derived from three climate models (i.e., the future climate conditions) with two representative scenarios demonstrated a decreasing baseflow trend in this basin, reaching a strong decreasing trend approximately in 2040. For the historical periods, the baseflow had two intersects using the MK test, showing that the response of baseflow was much more sensitive than that of streamflow. Streamflow flow lagged about 20 years behind baseflow. Annual baseflow anomalies are an efficient tool that can be used to evaluate drought events under future climate conditions. Our study predicted baseflow droughts (~10 years) in this catchment starting in 2041. Although it is challenging to forecast water-storage variations accurately (e.g., drought events), the baseflow projection from climatic scenarios in GCMs is a promising way to assess baseflow responses to future climatic changes.

**Author Contributions:** Conceptualization, J.Z. and P.Z.; methodology, P.Z., Y.Z., L.C., G.F. and G.Z.; software, P.Z.; validation, J.Z.; formal analysis, J.Z. and P.Z.; investigation, S.L.; resources, C.H. and M.M.; data curation, M.M.; writing—original draft preparation, J.Z. and P.Z.; writing—review and editing, J.Z., P.Z., Y.Z., L.C., G.F., Y.W., Q.L., S.L., S.Q., C.H. and G.Z.; visualization, J.Z.; supervision, J.S.; project administration, J.Z. and P.Z.; funding acquisition, P.Z. and G.Z. All authors have read and agreed to the published version of the manuscript.

**Funding:** This work was supported by the National Natural Science Foundation of China (42001034, 42101038, 41701022, 51679200 and 42207100), the Key Research and Development Program of Shaanxi (2019ZDLSF05-02), the Project for Outstanding Youth Innovation Team in the Universities of Shandong Province (2019KJH011), the Natural Science Foundation of the Shandong Province (ZR2019BD059), Postdoctoral Funding of China (2018M642692), the Second Tibetan Plateau Scientific Expedition and Research Program (STEP) (2019QZKK0903), and Key Science and Technology Program of Henan Province, China (222102320083).

**Data Availability Statement:** Data can be obtained upon request from the corresponding author.

**Acknowledgments:** We thank three anonymous reviewers and the editors for their thoughtful comments and suggestions. The authors would like to express their gratitude to EditSprings (https://www.editsprings.cn/, accessed on 3 September 2019) for the expert linguistic services provided.

**Conflicts of Interest:** The authors declare no conflict of interest.

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
