# Peer review of "Long-Term Baseflow Responses to Projected Climate Change in the Weihe River Basin, Loess Plateau, China"

_remotesensing, doi:10.3390/rs14205097_

Round 1

Reviewer 1 Report

Review for Remote Sensing – 1946721- Long-term baseflow responses to projected climate change in the Weihe River Basin, China

The title will be better: Long-term baseflow responses addressed to anthropogenic activities for projected climate change in the Weihe River Basin, China

The term “anthropogenic activities” is necessary due to increased population, agricultural production, industries, and related developmental activities (Chang et al., 2015). Land-use changes, as wrote the authors, are mainly due to the significant ecological plan.

Line 67 excludes the word: importantly, it is not necessary;

Line 85 (variability to baseflow were much more significant than those of forest disturbance) excludes the word: much, or rewrite.

Line 230. The SWAT hydrological model is a continuous-time, physically-based, computationally efficient, and semi-distributed catchment-scale hydrologic model (Jha et al., 2007).

The authors must exclude physically-based because the Swat model joins the USLE, an entirely empiric model to estimate soil losses.

            As the nonparametric Mann-Kendall test was used for trend analyses, the authors have to include the statistic (S and Z) of the Mann-Kendall test in the material and methods and results, which means including tables with Z values for each variable.

 Results/discussion and conclusion:

Include this adjustment below in the text in the abstract, results, discussion, and conclusion.

The baseflow showed a decreasing trend in some simulations of future climatic conditions but not in all scenarios.

The text in the results is concise; please improve and write more. With the results improved, the discussion and conclusions can also be improved.

Author Response

The title will be better: Long-term baseflow responses addressed to anthropogenic activities for projected climate change in the Weihe River Basin, China

The term “anthropogenic activities” is necessary due to increased population, agricultural production, industries, and related developmental activities (Chang et al., 2015). Land-use changes, as wrote the authors, are mainly due to the significant ecological plan.

R: Thank you for your constructive comments. The anthropogenic activities are the important derivers to influence the hydrological processes (e.g., baseflow) through the influence of hydro-meteorological variations (i.e., temperature and precipitation). We used the outputs from GCMs to project the baseflow in the future period. However, the meaning of anthropogenic activities is very broad. In this study, from the perspective of hydrology, we focused on the influence of the climate change on baseflow variations. Thus, in the context of changing climate, the climate change herein is contained the influence derived from anthropogenic activities. Therefore, the role of anthropogenic activities didn’t single out to highlight in the article title.

Line 67 excludes the word: importantly, it is not necessary.

R: Thanks for your comment. This word has been deleted in the Track Changes version.

Line 85 (variability to baseflow were much more significant than those of forest disturbance) excludes the word: much, or rewrite.

R: Thanks for your constructive comment. To clear this point, we rewrite the sentence follow your suggestion. The detailed were:

“Li, et al. [21] investigated the response of baseflow to climate variability in a large forested catchment and found that the contribution of climate variability on annual baseflow were greater than the impacts from forest disturbance.” in lines 66-68.

Reference:

  1. Li Q; Wei X; Zhang M; Liu W; Giles-Hansen K; Wang Y. The cumulative effects of forest disturbance and climate variability on streamflow components in a large forest-dominated watershed [J]. J Hydrol 2018, 557: 448-59.

Line 230. The SWAT hydrological model is a continuous-time, physically-based, computationally efficient, and semi-distributed catchment-scale hydrologic model (Jha et al., 2007).

The authors must exclude physically-based because the Swat model joins the USLE, an entirely empiric model to estimate soil losses.

R: Thanks for your constructive comment. We have deleted this description in the revised version. The details were:

“The SWAT hydrological model is a continuous-time, computationally efficient, and semi-distributed catchment-scale hydrologic model (Jha et al., 2007)” in lines 217-218.

As the nonparametric Mann-Kendall test was used for trend analyses, the authors have to include the statistic (S and Z) of the Mann-Kendall test in the material and methods and results, which means including tables with Z values for each variable.

R: Thanks for your constructive comment. The description of S and Z values have been added to method section 3.4. Figures were updated. The details were:

“The Z (derived from a certain climate element sequence) and S are trend and order column, and used to detect the significance test.” in lines 239-240.

Results/discussion and conclusion:

Include this adjustment below in the text in the abstract, results, discussion, and conclusion.

The baseflow showed a decreasing trend in some simulations of future climatic conditions but not in all scenarios.

R: Thanks for your constructive comment. We have rewritten the corresponding description in the abstract section. The details were:

“(1) that baseflow showed a decreasing trend in some simulations of future climatic conditions but not in all scenarios (p < 0.05)” in lines 34-35.

The text in the results is concise; please improve and write more. With the results improved, the discussion and conclusions can also be improved.

R: Thanks for your positive and constructive comments. We added more information to clear and improve the relevant content. The details were:

“The meteorological outputs in the GCMs were extracted as the inputs to SWAT model to predict the streamflow in the future climate change, and then, a well-revised Lyne-Hollick method was used to implement the baseflow separation.” in lines 254-256.

“The K and α are ranged from 62 ± 6 days (SD) and 0.98 ± 0.002 (SD) 1/day in the future two climate scenarios, respectively.” in lines 262-263.

“From the perspective of changing points, there demonstrated a similar and/or general pattern over a long future period, nevertheless the numbers of changing points were differing.” in lines 277-279.

“The outcomes of this method were more reproducible than the traditional methods (e.g., graphical approaches and empirical function [40]),” in lines 307-308.

“and detecting trend characteristics was beneficial to understanding the hydrological variability at a long-term scale” in lines 324-325.

“which reflects the holistic attribute of baseflow and terrestrial water balance [73]” in lines 352-353.

“and would be influenced by a variety of catchment physical factors [73]” in lines 377-378.

References:

  1. Singh S K; Pahlow M; Booker D J; Shankar U; Chamorro A. Towards baseflow index characterisation at national scale in New Zealand [J]. J Hydrol 2019, 568: 646-57.

Reviewer 2 Report

This paper dealt with an important topic and presents an interesting study on the changes and future projections of baseflow in the largest tributary river of the Yellow River. Generally, the paper is well organized and well written. The methods are sound and the results are quite clear. I recommend the paper be accepted after minor revision.

Comments

1. How the land cover data is acquired for the historical and future scenarios?

2. Lines 207-208, the meaning of this sentence is not clear, please rephrase.

3. Lines 100-102, these sentences should be moved to the Data/Method section.

Author Response

Comments and Suggestions for Authors

This paper dealt with an important topic and presents an interesting study on the changes and future projections of baseflow in the largest tributary river of the Yellow River. Generally, the paper is well organized and well written. The methods are sound and the results are quite clear. I recommend the paper be accepted after minor revision.

R: Thanks for your encouraging and positive comments. We have conducted the revision point-by-point followed your comments.

Comments

  1. How the land cover data is acquired for the historical and future scenarios?

R: Thanks for your comments. Actually, the land caver data wasn’t directly used in this study, and it only adopt to describe the status of cultivated land in the Weihe River in lines 138-139. Thus, we didn’t show the more information about the land cover data.

  1. Lines 207-208, the meaning of this sentence is not clear, please rephrase.

R: Thanks for your constructive comments. We have rephrased the sentence and make is clearer. The details were:

“On the context of climate change, three GCMs (i.e., the dry, moderate, and wet effects) were chosen to represent the future climate conditions.” in lines 191-192.

  1. Lines 100-102, these sentences should be moved to the Data/Method section.

R: Thanks for your constructive comments. We have moved them to the updated position. The details were:

“The General Circulation Model (GCM) is widely used to estimate the impacts of future climate conditions on hydrological cycles [24-28]” in lines 183-183.

Reviewer 3 Report

The authors took up an interesting and important problem regarding the predicted changes in the basal flow in one of the Chinese rivers. The paper was prepared correctly, appropriate research methods were used. The layout of the article is correct and the inference is correct. The authors made an extensive literature review.

  The weak point of the journal is the editorial side. The first errors and inaccuracies already appear in the list of authors, namely there is no attribution of the institution to a specific author.

The article was not prepared in accordance with the journal's guidelines. No line numbering. Citations should be numbered, not alphabetical, and arranged according to the order in which they are cited in the text. There is no need to duplicate figures and tables and present them in the text and after the list of references

Author Response

Comments and Suggestions for Authors

The authors took up an interesting and important problem regarding the predicted changes in the basal flow in one of the Chinese rivers. The paper was prepared correctly, appropriate research methods were used. The layout of the article is correct and the inference is correct. The authors made an extensive literature review.

The weak point of the journal is the editorial side. The first errors and inaccuracies already appear in the list of authors, namely there is no attribution of the institution to a specific author.

The article was not prepared in accordance with the journal's guidelines. No line numbering. Citations should be numbered, not alphabetical, and arranged according to the order in which they are cited in the text. There is no need to duplicate figures and tables and present them in the text and after the list of references.

R: Thank you for your encouraging and positive comments. We have conducted a large number of works to make the format to meet the publication of the Remote Sensing, e.g., added the line number, updated the format of the references, deleted the Figures and Tables listed the text. The corresponding content also have been improved by the other comments. The details were shown in the Track Changes version.

Round 2

Reviewer 1 Report

Dear authors

Stil is necessary to fix the text, as I asked before for the text to be published.